Rnf32 is not essential for spermatogenesis and male fertility in mice

Kong Hao
Yin Yufeng
Zeng Ni
Zhu Yunfei yunfei_zhu1006@126.com
Cui Yiqiang cuiyiqiang@126.com
State Key Laboratory of Reproductive Medicine and Offspring Health, Nanjing Medical University , Nanjing , China
Oliveira Sonia
Electronic publication date: 2025 Jul 30
Publication date: 2025
Volume: 13
Electronic Location ID: e19794
Received 2025 Mar 25; Accepted 2025 Jul 4
Copyright: ©2025 Kong et al.
Copyright year: 2025
Copyright holder: Kong et al.
License: This is an open access article distributed under the terms of the Creative Commons Attribution License, which permits unrestricted use, distribution, reproduction and adaptation in any medium and for any purpose provided that it is properly attributed. For attribution, the original author(s), title, publication source (PeerJ) and either DOI or URL of the article must be cited.
License URL: https://creativecommons.org/licenses/by/4.0/

Keywords: Rnf32, Spermatogenesis, Fertility, Testis, CRISPR/Cas9, Mice

Funding: National Key R&D Program of China 2022YFC2702800 This work was funded by the National Key R&D Program of China (2022YFC2702800 to Yiqiang Cui). The funders had no role in study design, data collection and analysis, decision to publish, or preparation of the manuscript.

==============================
Background

Ring finger motifs are found in a variety of proteins with diverse functions, often involved in protein-DNA or protein–protein interactions. The Rnf32-encoded protein contains two such motifs and is predominantly expressed in the testes and ovaries, suggesting that its expression may be regulated by elements within the Rnf32 promoter region. Rnf32 is active during spermatogenesis, mainly in spermatocytes and spermatids, indicating a potential role in sperm development.

Methods

We established an Rnf32 knockout (Rnf32−/−) mouse model using CRISPR/Cas9 technology. Gene expression was analyzed via reverse transcription quantitative polymerase chain reaction (RT-qPCR). Testicular and epididymal phenotypes were assessed through histological and immunofluorescence staining, and fertility and sperm motility were evaluated.

Results

Here, we successfully established an Rnf32 knockout mouse model using CRISPR/Cas9 technology. Surprisingly, male Rnf32−/− mice exhibited normal fertility, with no significant differences in testicular and epididymal histology, spermatogenesis, sperm count, or motility compared to Rnf32+/+ mice. These findings suggest that Rnf32 may not be essential for male fertility in mice, and its potential functions warrant further investigation.

Introduction

Recent epidemiological studies indicate that infertility affects nearly 15% of reproductive-age couples in China, with a persistent upward trend observed over the past two decades (De Kretser & Baker, 1999; Hackstein, Hochstenbach & Pearson, 2000). Male-related factors account for over 50% of these cases, predominantly linked to disrupted spermatogenic processes (Barratt et al., 2017). Clinically, such dysregulation manifests as compromised sperm parameters, including reduced counts, impaired motility, and morphological defects (McLachlan et al., 2007). Therefore, understanding the molecular mechanisms governing germ cell development represents a critical frontier in addressing male reproductive disorders. Spermatogenesis in mammals such as mice is a highly coordinated process of germ cell development, predominantly occurs in the seminiferous tubules through three key phases: spermatogonial stem cell self-renewal and differentiation, meiotic division of spermatocytes, and post-meiotic spermiogenesis involving cytoplasmic remodeling (Chen et al., 2018). Mature sperm released into the tubular lumen undergo functional maturation during epididymal transit, acquiring progressive motility and fertilization competence (Cooper, 2007; Dacheux & Dacheux, 2014). Final activation events, including capacitation and acrosome reaction, occur in the female reproductive tract to enable successful oocyte fusion (Ickowicz, Finkelstein & Breitbart, 2012; Saint-Dizier et al., 2020).

RPL39L, a germline-specific ribosomal protein, exhibits stage-specific expression in pachytene spermatocytes and post-meiotic germ cells (round/elongated spermatids). Phenotypic analysis of RPL39L-deficient mice revealed impaired sperm quality parameters (morphology, count, and motility), confirming its essential role in male reproductive competence. Proteomic analysis further revealed that the silencing of RPL39L induces the coordinated downregulation of 276 cellular proteins (Zou et al., 2021; Li et al., 2022). Among these downregulated proteins, we specifically focused on Rnf32, hypothesizing that it may also be a component of the ribosomal or translational regulatory network and is specifically expressed in the testis. Given its potential role and testis-specific expression, we propose that Rnf32, like RPL39L, may play a critical role in male reproduction. This chromosome 7q36-located gene encodes a 362-amino acid protein containing two RING-H2 zinc-binding domains flanking a central IQ calmodulin-binding motif. Notably, Rnf32 represents the first reported gene encoding a dual RING-H2 domain protein in mammals (Van Baren et al., 2002), positioning it as a unique model for investigating ubiquitination-related regulatory networks in reproductive biology.

Transcript analysis of Rnf32 splice variants revealed distinct functional impacts: Exon 1 variations maintain the open reading frame (ORF) due to the initiation codon’s exclusive localization in exon 2. However, a premature termination codon in exon 3a causes protein truncation upstream of the first RING domain. Furthermore, extended exon 4 transcripts contain a 16-base termination signal immediately downstream of the canonical splice site, resulting in ORF truncation within the first RING-H2 domain and producing a 145-amino acid polypeptide (Van Baren et al., 2002; Chasapis & Spyroulias, 2009). Tissue-specific expression profiling demonstrated Rnf32’s predominant transcription in gonadal tissues, with stage-specific upregulation during spermatogenesis. This spatiotemporal expression pattern, particularly in meiotic spermatocytes and post-meiotic germ cells, strongly suggests its functional involvement in gamete maturation (Yu et al., 2003; Guo et al., 2004). Despite our existing understanding of Rnf32, its specific role in spermatogenesis remains enigmatic. To further explore this issue, this study adopted the advanced CRISPR/Cas9 technology and successfully constructed a Rnf32 gene knockout mouse model. With the help of this model, we hope to unambiguously reveal the physiological role played by Rnf32 in the complex mechanism of spermatogenesis.

Materials & Methods

Animal experiments

All experimental animals were obtained from the Animal Core Facility of Nanjing Medical University. The animal experimentation involving C57BL/6 mice adhered to the guidelines established by the Institutional Animal Care and Use Committee (IACUC) of Nanjing Medical University (IACUC-2307030). The experiments were conducted under specific pathogen-free (SPF) conditions, with a light cycle of 12/12 h, a relative humidity ranging from 50 to 55%, and an ambient temperature maintained at (23 ± 2) °C. Throughout the study, the animals had unrestricted access to autoclaved feed and sterile water, and humane endpoints were strictly observed. The mice were randomly assigned to cages, with each cage accommodating 4–5 mice. All cages were maintained under similar conditions, including cage density, bedding, and cleaning frequency. To harvest testicular and epididymal tissues from adult mice for subsequent experimental analyses, our procedures adhered to the 2020 edition of the American Veterinary Medical Association’s Guidelines for the Euthanasia of Animals. Euthanasia was performed using compressed carbon dioxide (CO2) delivered into a rodent euthanasia chamber. The CO2 flow rate was calibrated to displace 30%–70% of the chamber volume per minute, ensuring a consistent and controlled introduction of the gas. This method rapidly induced unconsciousness in the animals, thereby minimizing potential distress. Following the cessation of breathing, CO2 administration was maintained for at least one additional minute to confirm euthanasia. This approach aligns with established protocols in life sciences research, prioritizing both ethical considerations and experimental reproducibility. The Rnf32 knockout model (C57BL/6N background) was established through CRISPR/Cas9-mediated embryonic genome editing. Two sgRNAs (sgRNA-1: 5′-gatattggttctttctacagagg-3′; sgRNA-2: 5′-aacaagagtgatacatgatgggg-3′) were co-delivered with Cas9 mRNA into zygotes via microinjection. We initially introduced Cas9 and sgRNA into embryos to obtain F0 generation mice. Through subsequent breeding for three generations, we successfully established a stable knockout mouse line.

RNA extraction and quantitative real-time PCR

Total RNA isolation from Rnf32+/+ (n = 3) and Rnf32−/− (n = 3) murine specimens was performed using TRIzol reagent (15596026; Thermo Fisher Scientific) following standard phenol-chloroform extraction protocols. Complementary DNA (cDNA) synthesis was carried out with the PrimeScript™ Reverse Transcriptase kit (RR037A; Takara Bio) according to the manufacturer’s protocol. The resultant cDNA served as template for both conventional reverse transcription-PCR (RT-PCR) using 2 × RapidTaq Master Mix (P515; Vazyme Biotech) and quantitative real-time PCR (RT-qPCR) with AceQ™ SYBR Green Master Mix (Q131; Vazyme Biotech). Amplification was conducted on a QuantStudio 5 Real-Time PCR system (A28573; Applied Biosystems) under optimized thermal cycling parameters: initial denaturation at 95 °C for 10 min, followed by 40 cycles of denaturation (95 °C, 10 s), annealing (65 °C, 1 min), signal acquisition (97 °C, 1 s), and final extension (37 °C, 30 s). In all reverse transcription quantitative polymerase chain reaction (RT-qPCR) assays, 18S ribosomal RNA (rRNA) was employed as the endogenous control to ensure accurate normalization of gene expression data. Primers were obtained from PrimerBank (ID:31981166a1), which provides comprehensive validation data including QPCR Validation Results, Amplification Plots, and Dissociation Curves. Primer sequences are detailed in Table S1.

Fertility test

To evaluate reproductive competence, C57BL/6J adult male mice (8–10 weeks old) of Rnf32+/+ and Rnf32−/− genotypes (n = 3 per group) were housed in monogamous mating pairs with three sexually mature Rnf32+/+ females (postnatal day 56). A restricted 12-hour nocturnal mating window was implemented, during which individual male–female pairs were cohabited. Copulatory plug formation was verified post-coitum as a primary fertilization indicator, followed by longitudinal tracking of parturition outcomes to quantify litter size. This experimental paradigm was triplicated using independent male cohorts to ensure statistical robustness.

Harvesting of tissues and histological analysis

Testicular and epididymal tissues were promptly dissected following euthanasia of the adult male mice (8–10 weeks old). Specimens underwent fixation in modified MDF fixative (24 h minimum), followed by graded ethanol dehydration and paraffin embedding. Serial sections (five µm thickness) were prepared using a rotary microtome and mounted on poly-L-lysine-coated slides. Deparaffinized sections were subjected to hematoxylin and eosin (H&E) staining for morphological assessment using standardized histopathological protocols. All tissues collected from unmated males.

Sperm motility and sperm count assays

Following euthanasia of the mice, epididymal tissues were microsurgically dissected from adult C57BL/6 mice (postnatal day 56; n = 3) and immediately immersed in modified human tubal fluid (HTF) medium (#90126; Irvine Scientific) supplemented with 10% fetal bovine serum. The cauda epididymis was incised under sterile conditions to release spermatozoa into the medium, followed by a 5-min incubation at 37 °C. Computer-assisted sperm analysis (CASA; Hamilton Thorne IVOS II) was subsequently performed to quantify sperm concentration and kinematic parameters. All tissues collected from unmated males.

Immunofluorescence

Paraffin-embedded testicular sections (five µm thickness) underwent sequential deparaffinization in xylene, rehydration through graded ethanol series, and heat-mediated antigen retrieval in 10 mM sodium citrate buffer (pH 6.0) at 95 °C for 15 min. Non-specific binding was blocked with 5% BSA in PBS (2 h, RT) prior to overnight incubation with primary antibodies at 4 °C. After three PBST (0.05% Tween-20) washes, sections were incubated with Alexa Fluor-conjugated secondary antibodies (2 h, RT) and counterstained with Hoechst 33342 nuclear dye (Invitrogen; 5 min). Fluorescent images were acquired using a Zeiss LSM800 confocal system with ZEN imaging software.

TUNEL

Apoptotic activity was evaluated using terminal deoxynucleotidyl transferase dUTP nick-end labeling (TUNEL) assay (Cat# A112; Vazyme Biotech) according to standardized protocols. Paraffin-embedded testicular sections (five µm thickness) underwent sequential processing including rehydration, proteinase K digestion (20 µg/ml, 15 min), and enzymatic labeling with TUNEL reaction mixture (37 °C, 1 h). Fluorescence imaging was performed using a Zeiss LSM 880 confocal system (Oberkochen, Germany) equipped with Airyscan detection. Quantitative assessment of germ cell apoptosis was conducted by systematically analyzing fifty seminiferous tubule cross-sections per biological replicate, with apoptotic index calculated as TUNEL-positive cells per tubular circumference.

Statistical analysis

Quantitative data are presented as mean ± SEM. Statistical analyses of inter-genotypic differences were performed using two-tailed unpaired Student’s t-tests implemented in GraphPad Prism v9.0 (GraphPad Software, La Jolla, CA, USA). Significance thresholds were defined as follows: ∗P < 0.05, ∗∗P < 0.01, ∗∗∗P < 0.001, and ∗∗∗∗P < 0.0001, with “ns” indicating non-significance.

Results

The expression of Rnf32 and the generation of Rnf32−/− mice

Rnf32 is highly conserved in mammals (Van Baren et al., 2002). Therefore, studying its reproductive effects on mice provides critical insights into human reproductive pathologies. To explore its biological roles, we first performed quantitative PCR (qPCR) to analyze tissue-specific Rnf32 expression profiles in mice. This analysis identified predominant Rnf32 expression within the male reproductive system, with testicular tissue showing significantly higher transcript levels compared to other examined organs (Fig. 1A). Developmental time-course analysis revealed stage-specific upregulation, with Rnf32 expression peaking at postnatal week 3 (Fig. 1B), a critical period corresponding to spermatid differentiation during murine testicular maturation (Gong et al., 2013; Nishimura & L’Hernault, 2017; Chen et al., 2024). To mechanistically investigate Rnf32’s role in gametogenesis, we engineered a CRISPR/Cas9-mediated exon 7 in-frame deletion (Fig. 1C) through microinjection of sgRNA/Cas9 complexes into superovulated zygotes. Exon 7 was selected as the target site because it encodes a critical structure motif within the zinc finger domain in the Rnf32. By specifically targeting this domain for disruption, we have effectively compromised the structural integrity and consequently maximally impaired the functional capacity of this protein. Sanger sequencing confirmed successful generation of Rnf32−/− mice, revealing a precise 35-bp deletion within exon 7 (Fig. 1D). Transcript analysis via RT-PCR confirmed efficient Rnf32 knockout, demonstrating significantly reduced mRNA levels in testicular tissue from Rnf32−/− animals compared to wild-type controls (Fig. 1E).

Figure 1 The expression of Rnf32 and the generation of Rnf32−/− mice.

(A) Tissue-specific Rnf32 transcript profiling by RT-PCR analysis. mRNA levels were quantified across multiple murine organs, with data normalized to 18S rRNA endogenous control (n = 3). (B) Developmental-stage-specific expression profiles of Rnf32 in testicular tissue. Temporal dynamics were assessed at defined postnatal intervals (n = 3). (C) CRISPR-Cas9-mediated homologous recombination strategy for Rnf32 knockout. Schematic illustrates sgRNA target sites flanking exon 7. Electropherogram analysis verified a 35-bp frameshift deletion (red box) inducing a premature termination codon. (D) Validation of the Rnf32−/− allele by bidirectional Sanger sequencing. Chromatogram depicts precise genomic deletion (dashed line) in mutant mice. (E) Comparative transcript analysis of Rnf32 mRNA in wild-type versus Rnf32−/− testicular tissue, n = 3, ∗∗∗∗P < 0.0001.

Rnf32−/− mice are fertile

We conducted an examination of testicular morphology and testicular/body weight ratios in Rnf32-KO male mice, finding no statistically significant difference compared to the control group (Figs. 2E–2D). Following this, we carried out fertility tests to investigate the impact of Rnf32 on mouse fertility, counting the pups and litters in both the experimental and control groups. The findings indicated that there was no statistically significant disparity in fertility between the WT and KO mice (Fig. 2E). In addition, we performed H&E staining on the testis, revealing normal spermatogenic cells of all stages of adult (postnatal 8w) Rnf32−/− mice (Fig. 3A).

Figure 2 Rnf32−/− mice are fertile.

(A–C) Body weight, testis weight and testis/body weight ratio of adult Rnf32+/+ and Rnf32−/− mice, n = 3, P > 0.05. (D) Representative images of testes and epididymides of adult (postnatal day 56) Rnf32+/+ and Rnf32 −/− mice. (E) The litter size of Rnf32+/+ and Rnf32−/− mice, n = 3, P > 0.05.

Figure 3 Spermatogenesis is normal in Rnf32−/− mice.

(A) H&E staining of the testis from adult Rnf32+/+ and Rnf32−/− mice. (B) H&E staining of seminiferous tubules from adult Rnf32+/+ and Rnf32−/− mice. L, leptotene spermatocyte; Z, zygotene spermatocyte; P, pachytene spermatocyte; M, meiotic spermatocyte; D, diplotene spermatocyte; rSt, round spermatid; eSt, elongated spermatid.

The knockout of Rnf32 does not impair spermatogenesis

Spermatogenesis, the fundamental biological process of germ cell differentiation, proceeds through precisely coordinated developmental stages in murine testes. The seminiferous epithelial cycle is classically categorized into 12 distinct phases based on spermatid maturation (Abe, Shen & Takano, 1991; Hess & Renato De Franca, 2008; Griswold, 2016), as defined by Bouin’s fixative (picric acid/formaldehyde/acetic acid)-preserved tissue analysis coupled with periodic acid-Schiff (PAS)-hematoxylin staining for precise staging of seminiferous epithelial cycles. To evaluate potential spermatogenic disruptions in Rnf32+/+ and Rnf32−/− mice, we systematically analyzed germ cell cytoarchitecture across all 12 epithelial stages. Histological assessment revealed preserved seminiferous tubule organization in knockout testes, displaying complete germ cell lineages—including spermatogonia, spermatocytes, and elongating spermatids—with normal spatiotemporal progression of germ cell maturation indistinguishable from wild-type controls (Fig. 3B). Immunofluorescence quantification demonstrated comparable expression of SOX9, a Sertoli cell-specific transcription factor essential for spermatogonial niche maintenance, between adult Rnf32+/+ and Rnf32−/− mice (Figs. 4A–4B). Quantitative data can be found in the Figs. S1A–S1B. Furthermore, TUNEL assay revealed statistically comparable rates of luminal germ cell apoptosis between adult Rnf32 +/+ and Rnf32−/− mice (Figs. 4C–4D). Quantitative data can be found in the Figs. S1C–S1D.

Figure 4 The knockout of Rnf32 doesn’t affect spermatogenesis and apoptosis.

Sertoli cells (SOX9) and spermatids (PNA) is comparable in adult (A) Rnf32+/+ and (B) Rnf32−/− male mice testis. Apoptotic cells is comparable in adult (C) Rnf32+/+ and (D) Rnf32−/− male mice testis.

The knockout of Rnf32 has no impact on sperm counts, morphology and motility

To investigate the potential impact of Rnf32 deficiency on sperm maturation in the epididymis, histopathological evaluation of the cauda epididymidis was performed. Comparative analysis demonstrated conserved sperm concentration in Rnf32−/− epididymal luminal contents relative to wild-type controls (Fig. 5A). Ultrastructural assessment via differential interference contrast microscopy revealed preserved spermatozoal morphology in mutant mice, including typical head curvature and flagellar architecture (Fig. 5B). Quantitative data can be found in the (Fig. S1E). Computer-assisted semen analysis (CASA) was employed to quantify sperm kinematic parameters, such as count, motility, and progressive motility. The results indicated that there were no significant disparities in these metrics between Rnf32 +/+ and Rnf32−/− mice (Figs. 5C–5E).

Figure 5 The knockout of Rnf32 has no impact on sperm counts, morphology and motility.

(A) H&E staining of cauda epididymidis from adult Rnf32+/+ and Rnf32−/− mice. (B) H&E staining of sperm from adult Rnf32+/+ and Rnf32−/− mice. (C–E) Sperm count, sperm motile and sperm progressive were assessed in adult Rnf32+/+ and Rnf32−/− mice, n = 3, P > 0.05. All data are presented as means ± SEM, analyzed using a two-tailed unpaired t-test, ns indicates no difference.

Discussion

The utilization of genetically engineered mouse models (GEMMs) has profoundly enhanced our mechanistic understanding of reproductive developmental disorders associated with human infertility. Systematic interrogation of germline-specific gene ablation models has provided critical insights into the regulatory networks governing stem cell self-renewal, meiotic progression, and spermatogenic differentiation (Cooke & Saunders, 2002; Gilbert, Roof & Rajendra Kumar, 2018; Azhar et al., 2021). In this study, the generation of a CRISPR/Cas9-mediated Rnf32 knockout murine model was achieved, enabling comprehensive phenotypic characterization. Comparative analysis revealed no significant genotypic differences in reproductive competence between Rnf32−/− and wild-type (WT) males. These findings demonstrate that Rnf32 is dispensable for murine male fertility, aligning with emerging evidence that numerous testis-enriched genes exhibit functional redundancy in reproductive contexts (Lv et al., 2022; Chen et al., 2022; Meng et al., 2023). Importantly, this study precludes RNF32 as a viable contraceptive target or monogenic infertility factor, thereby optimizing research resource allocation by eliminating nonproductive investigative pathways.

While Rnf32 appears dispensable for baseline spermatogenesis, we hypothesize that Rnf32 may have a potential cytoprotective role in maintaining male fertility, particularly under conditions of environmental toxicity. Notably, certain genetic regulators, exemplified by the melanoma antigen (MAGE) family genes, demonstrate functional specialization in xenobiotic detoxification pathways. Although MAGE proteins are nonessential for physiological gametogenesis, they confer resilience against toxicant-induced spermatogenic impairment by modulating oxidative stress response cascades (Fon Tacer et al., 2019; Florke Gee et al., 2020; Li et al., 2023). Intriguingly, Rnf32 shares evolutionary conserved structural motifs with these xenobiotic-responsive factors, suggesting possible involvement in analogous molecular mechanisms. We postulate that Rnf32 may activate compensatory proteostatic networks under specific pathological conditions, thereby maintaining genomic integrity during spermatogonial stem cell proliferation and meiotic recombination. This hypothetical safeguard mechanism could explain the preserved fertility observed in Rnf32-deficient mice under controlled laboratory conditions while highlighting its potential clinical relevance in environmentally compromised reproductive scenarios. Another possibility is that functionally related genes may compensate for the loss of Rnf32 upon its knockout. As Rnf32 is a member of the extensive RING finger protein family, characterized by conserved E3 ubiquitin ligase activity, it is plausible that structurally homologous family members exhibiting spatiotemporal expression overlap with Rnf32 may functionally compensate for the absence of Rnf32 through redundant or complementary enzymatic mechanisms. While our study employed a multi-modal approach to evaluate reproductive function—including fertility metrics (litter size) and spermatogenesis endpoints (sperm count, motility, histopathology)—the limited cohort size (n = 3 per group) constrains statistical power and generalizability. Additionally, our observations were confined to male mice.

Conclusions

Despite robust Rnf32 expression in murine testicular tissue, genetic ablation of this locus failed to induce discernible morphological abnormalities in testicular dimensions or spermatogenic cell populations. Crucially, comparative analysis of breeding performance revealed preserved reproductive capacity in Rnf32-KO males relative to wild-type counterparts.

Supplemental Information

Supplemental Information 1 Primers used in PCR,RT-PCR and RT-qPCR

Supplemental Information 2 Raw CT values

Supplemental Information 3 (A) PNA positive apoptotic cells counts in adult Rnf32+/+ and Rnf32−/− mice, n = 3, P > 0.05

(B) SOX9 positive apoptotic cells counts in adult Rnf32+/+ and Rnf32−/− mice, n = 3, P > 0.05. (C) TUNEL positive apoptotic tubule counts in adult Rnf32+/+ and Rnf32−/− mice, n = 3, P > 0.05. (D) TUNEL positive apoptotic cells counts in adult Rnf32+/+ and Rnf32−/− mice, n = 3, P > 0.05. (E) Percentage of abnormal sperm in adult Rnf32 +/+ and Rnf32−/− mice; n = 3, P > 0.05.

Supplemental Information 4 Original Figs. 3D–3E

Supplemental Information 5 Raw data

Supplemental Information 6 Raw data analysis

Open with GraphPad Prism (https://www.graphpad-prism.cn).

Supplemental Information 7 Author checklist

Supplemental Information 8 MIQE checklist

We thank Jinyang Cai for continuous support with microscopy.

Additional Information and Declarations

Competing Interests

Author Contributions

Animal Ethics

Data Availability

The authors declare there are no competing interests.

Hao Kong conceived and designed the experiments, performed the experiments, analyzed the data, prepared figures and/or tables, and approved the final draft.

Yufeng Yin conceived and designed the experiments, performed the experiments, analyzed the data, prepared figures and/or tables, and approved the final draft.

Ni Zeng analyzed the data, prepared figures and/or tables, and approved the final draft.

Yunfei Zhu analyzed the data, authored or reviewed drafts of the article, and approved the final draft.

Yiqiang Cui analyzed the data, authored or reviewed drafts of the article, and approved the final draft.

The following information was supplied relating to ethical approvals (i.e., approving body and any reference numbers):

Institutional Animal Care and Use Committee of Nanjing Medical University.

The following information was supplied regarding data availability:

The raw measurements are available in the Supplementary File.

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
