# Peer review of "Rnf32 is not essential for spermatogenesis and male fertility in mice"

_PeerJ, doi:10.7717/peerj.19794_

## Round 0.1 · original submission · Major Revisions

Dear authors,

Thank you for your submission. While the study presents a valuable contribution to the understanding of Rnf32 in male fertility, several significant points raised by the reviewers must be addressed to ensure the scientific rigour and clarity.

- Please provide additional detail on the CRISPR/Cas9 strategy, particularly regarding the generation of knockout mice and rationale for targeting exon 7, as well as clarification of RT-qPCR normalisation procedures and primer efficiency validation.

- In-line citations must be reviewed and corrected. Ensure that appropriate references support each claim.

- Where qualitative data are presented (e.g., fluorescence, sperm morphology), include representative images from replicates or quantitative metrics where feasible.

- Expand on the potential redundancy among E3 ligases and place Rnf32 in context with related fertility genes to strengthen your conclusions.
- Please attend to formatting conventions (e.g., italicisation of gene vs protein names), clarify species-specific findings, and ensure all supplementary materials are complete (e.g., genotyping primers).

Please, refer to the reviewers' comments for further details. We look forward to receiving a thoroughly revised manuscript.

Reviewer 1 ·

Basic reporting

This manuscript demonstrates that Rnf32 is not essential for infertility via Rnf32 knockout mice.
It appears that the authors may no longer intend to investigate this mouse further; however, could you provide data on Rnf32 protein expression in the knockout mice and controls using immunohistochemistry or Western blot? If protein expression analysis is not possible, please provide a clear explanation for this limitation. In Figures 3 B-C, the inclusion of IF markers specific to spermatogenesis and fertility findings is relevant, but it would be important to report the percentage of marker-positive cells relative to the total number of evaluated cells. Finally, the authors should include "mice" in the keywords.

Experimental design

-

Validity of the findings

-

Reviewer 2 ·

Basic reporting

Some of the in-line citations appear to be in the wrong place, and there are additional places in the manuscript where citations should be added. See main comments, comment #3, for more information and specific examples.

Experimental design

Some of the methods require more information, specifically related to how the knockout mice were generated and how the RT-qPCR data was normalized and analyzed. See main comments #1 and #2 for details.

Validity of the findings

No comment

Additional comments

Kong et al. investigated the role of Rnf32 in mouse spermatogenesis. This gene shows testis-specific expression in mice, but it is unclear if it is essential for male fertility or what role it may play in spermatogenesis. To test this, the authors generated Rnf32 knockout mice using CRISPR/Cas9 and showed that, surprisingly, Rnf32 does not appear to be essential for male fertility in mice. The experimental approaches are sound, and I followed the paper very clearly. However, there is additional information I would like to see in the methods to be able to fully evaluate the paper. Specifically, I would like more information about how the knockout mice were generated and how the RT-qPCR results were normalized. If the authors’ methods in these two areas are sound, then I think this work will represent a rigorous investigation into the role of Rnf32 in mouse male fertility.

Major Comments:
1. Please include more information about how the CRISPR/Cas9 knockout mice were generated. Were the injectants considered homozygous knockouts, and if so, is it possible there were different mutations in different mice used in this study? Or did the authors breed injected mice to establish a stable line?
2. Please also provide more information about the RT-qPCR normalization and validation. The figure 1 legend says that tissue-specific expression profiling of Rnf32 used an 18S rRNA control, but the methods do not mention this. The relative expression level of Rnf32 in testis is 1, and I am not sure how the value can be exactly 1 if it was normalized to the expression level of another gene. It is also not clear to me if all RT-qPCR experiments were normalized using 18S rRNA (data in figures 1B and 1E). Also, did the authors do any tests to validate their primer efficiency?
3. There are issues with the in-line citations. For example, van Baren et al. (2002) seems to be the main paper establishing Rnf32 expression in testes, and it is listed in the references (#18), but I do not see it cited anywhere in the main text. The sentence that talks about RNF32 structure cites Nishimura et al., 2002 and Tanaka et al., 2007 (line 57), but these papers are both about spermatogenesis expression, so they do not seem like the appropriate citations for this sentence. There are also sentences throughout the manuscript that should have citations but do not. For example, in line 169, what is the evidence that Rnf32 is conserved across species? There are similar examples in lines 176 and 195-200.
4. Results that are supported by images should have additional images or quantitative data to back them up. The authors did this well with testes mass, showing the representative images in Figure 2D and quantitative data in Figure 2B. For results involving fluorescent labeling or sperm morphology, are there quantitative metrics the authors can provide? For example, could the authors calculate relative fluorescence or percent abnormal sperm with the data they have? If not, please provide images from biological replicates in the supplement to show that the images in the main text are representative of all samples.

Minor Comments:
1. In the introduction, is this background information from mice, humans, or a combination of both? Clarify which systems these previous results come from.
2. Rnf32 should be italicized throughout the text when using the mouse gene symbol: https://www.informatics.jax.org/mgihome/nomen/short_gene.shtml
3. The sample size is a bit small (n=3). I understand that larger sample sizes are probably not feasible in mice, and I do not think low power is the reason for non-significant results since the results are not even trending towards any evidence for reduced fertility. But the caveat of sample size could be mentioned in the discussion.
4. The authors should include information in the methods about tissue and developmental timepoint sampling. How were the data collected for figures 1A and 1B?
5. Were tissues and sperm collected from unmated males for phenotype and histology analyses? This should be addressed in the methods.
6. I like the hypothesis in the discussion that Rnf32 may be important for spermatogenesis under pathogenic or toxic conditions. The authors might also consider that Rnf32 could be important for sperm competition and cryptic female choice, or that similar genes could compensate when Rnf32 is knocked out.
7. Check line numbers in the MIQE checklist

Reviewer 3 ·

Basic reporting

The manuscript is clear and well written. The main literature is cited, but the authors may consider to expand the discussion by adding some sentences on other genes that have been proved essential for male fertility (e.g., Inoue et al. Nature 2005, Barbaux et al. Sci Rep 2020, Lamas-Toranzo et al. eLife 2020…).

Experimental design

The research is original as no KO model was available for this gene. The authors explore the role of the protein Rnf32 on mice fertility following a rationale well explained in the introduction and further substantiated by Rnf32 expression patterns show in Figure 1. The fertility test on Fig. 2E would suffice to prove the essential role of Rnf32, but the manuscript also provides other histological analyses and sperm functionality tests to prove that the ablation did not cause subtle alterations in spermatogenesis. Methods are well described.

Validity of the findings

I consider that the manuscript provides a clear answer on the dispensable role of Rnf32 on male fertility. The finding is relevant and advance in the current knowledge of genes involved in male reproduction.

Additional comments

Line 69: Add space “2003). Despite”.
Line 194: Change to “does not impair”.
Figure 3D: It would be advisable to provide the original picture (i.e., without the 35 bp) of the -/- sequence below.

Reviewer 4 ·

Basic reporting

The manuscript is clearly written in professional and grammatically correct English, with a logical flow. The introduction places the study in context, providing relevant background on the Rnf32 gene, its expression pattern in gonadal tissues, and its hypothesized role in spermatogenesis. The references cited are appropriate and up-to-date, and help to justify the rationale for generating a knockout model. The figures are relevant, well-labeled, and appropriately described. The raw data files and supplementary materials are provided and appear to be sufficient and support the claims made in the text.

The manuscript would benefit from additional references in places where prior findings are described. For example, Lines 33–34 (Male-related factors), Lines 54–55 (Chromosome 7q36-located gene), Lines 225–227 (Regulatory networks governing stem cell self-renewal, meiotic progression), Lines 237-239 (cryoprotective functions), and Lines 243-245 (Rnf32 shares evolutionary conserved structural motifs) would each be strengthened by the addition of appropriate citations to relevant literature.

Lines 50–58 introduce RPL39L as a ribosome-related fertility factor and then transition to Rnf32.
However, the functional relationship between the two is not clearly established. It would be helpful to clarify whether Rnf32 is also considered part of the ribosome or translational regulation network, or whether it represents a mechanistically distinct category (for instance, ubiquitination pathway). Clarifying the group of genes or pathways Rnf32 represents would strengthen the rationale for its selection and highlight its potential significance in reproductive biology.

Experimental design

The research question is clearly defined; whether Rnf32 is required for normal spermatogenesis and fertility in mice. The authors address this question with appropriate assays, including RT-qPCR to confirm gene knockout, fertility testing via litter size analysis, histological and immunofluorescence examination of testes and epididymides, and TUNEL assays for apoptosis. Each experiment is well described, and methods are sufficiently detailed to enable replication. Sample sizes are modest but consistent with standard practices in mouse fertility studies (n=3 per group), though additional biological replicates could improve statistical robustness.

The authors should consider briefly discussing the evolutionary conservation of Rnf32, particularly across mammals, to highlight its biological importance and the relevance of the mouse model for human reproductive biology.

One point that would benefit from clarification is the choice of exon 7 for CRISPR targeting. While the deletion strategy is clearly described and appears effective, the rationale for selecting exon 7, rather than targeting an earlier exon, is not provided. It would strengthen the manuscript to briefly explain whether exon 7 encodes an essential protein domain (for example, a RING finger motif), affects all known isoforms, or introduces a frameshift leading to loss of function.

To enhance reproducibility, the authors should provide the primer sequences that were used to detect the 35 bp deletion in the Rnf32 mutant allele. While genotyping results are shown, the actual primer sequences are not included in the Supplementary Table 1. Adding these to the Supplementary Materials would assist readers in replicating the genotyping strategy used in this study.

In Figure 1E, it would improve transparency and reproducibility if the authors included individual Ct curves or raw Ct values for each biological replicate. This would allow readers to assess amplification specificity and variability across replicates, in line with best practices for RT-qPCR.

In Figures 3D-E, adding arrows to highlight the apoptotic regions would enhance clarity for readers. This would help emphasize the TUNEL-positive signals and support the interpretation of apoptosis regions.

Validity of the findings

The data are statistically sound. The authors demonstrate that Rnf32 knockout mice exhibit no significant differences in testis morphology, sperm parameters, or fertility outcomes compared to wild type controls. The histology and apoptosis analyses further support the conclusion that Rnf32 is not essential for normal spermatogenesis under baseline conditions.

However, the manuscript would benefit from a brief discussion of potential functional redundancy. Since Rnf32 belongs to a large family of RING finger proteins, it is possible that other E3 ubiquitin ligases with overlapping expression compensate for its loss. Including whether known paralogs or related genes are expressed in the testis would provide valuable context for interpreting the negative phenotype and guide future studies.

---

## Round 0.2 · Minor Revisions

Dear authors,

There are only some minor revisions suggested by the reviewers before publication. Please, refer to their reports. (note: do not forget to proofread your references section / citations)

Reviewer 1 ·

Basic reporting

The authors have done a good job of revision, and the paper meets the publication requirements

Experimental design

no comment

Validity of the findings

no comment

Reviewer 2 ·

Basic reporting

The authors did a thorough job addressing most of my comments, and I greatly appreciate the effort they put into this revision. The references look better, but there are still a few places where the authors should double check their references. For example, line 34 is discussing human infertility and cites Chen et al. 2018. I am not familiar with this paper, but it appears to be about mice based on the title. Also, some papers listed under “references” do not appear to be cited at all in the main text, such as #1 (De Kretser and Baker 1999) and #3 (Barratt et al. 2017).

Experimental design

The authors did an excellent job answering my questions about the knockout mice and RT-qPCR analyses. I encourage the authors to put this information in the main text, in addition to the reviewer responses. Specifically, I think the authors should state that they generated stable knockout lines with 3 generations of breeding under the “Animal experiments” section in the methods. They should also give the accession number for their primers and state that they have already been validated in the “RNA extraction and quantitative real-time PCR” section in the methods.

Validity of the findings

The authors did a very thorough job and checked many phenotypes related to fertility, which gives a very convincing argument that Rnf32 is not essential for male fertility in mice.

Additional comments

Thank you for this thorough revision and for sufficiently addressing all of my methodological concerns. I commend the authors on this very rigorous study.

Reviewer 4 ·

Basic reporting

-

Experimental design

I thank authors making the necessary changes.

I think it would be useful to:
1) expand the section on the evolutionary conservation of Rnf32, particularly across mammals, to highlight its biological importance and the relevance of the mouse model for human reproductive biology (lines 173-174 in the revised pdf)
2) add the reasoning of choosing exon 7 by CRISPR targeting in the manuscript

Validity of the findings

-

---

## Round 0.3 · accepted · Accept

Dear authors,
congratulations! I am now accepting your manuscript for publication. Many thanks for your hardwork.